# Influence of Sb^3+^ Cations on the Structural, Magnetic and Electrical Properties of AlFeO_3_ Multiferroic Perovskite with Humidity Sensors Applicative Characteristics

**DOI:** 10.3390/ma15238369

**Published:** 2022-11-24

**Authors:** Iulian Petrila, Florin Tudorache

**Affiliations:** 1Faculty of Automatic Control and Computer Engineering, Gheorghe Asachi Technical University of Iasi, Str. Dimitrie Mangeron, no. 27, 700050 Iasi, Romania; 2Institute of Interdisciplinary Research, Department of Exact and Natural Sciences, Ramtech Center, Alexandru Ioan Cuza University of Iasi, Boulevard Carol I, no. 11, 700506 Iasi, Romania

**Keywords:** perovskites, electrical properties, magnetic properties, humidity sensor

## Abstract

The effects of Sb^3+^ cations substitution on the structural, magnetic and electrical properties of Al_1−x_Sb_x_FeO_3_ multiferroic perovskite are investigated. The partial or total substitution of Al^3+^ cations with Sb^3+^ cations, in stoichiometric composition Al_1−x_Sb_x_FeO_3_ (x = 0.00, 0.25, 0.50, 0.75 and 1.00) were made in order to identify composite materials with sensors applicative properties. Multiferroic perovskite samples were prepared following technology of the ceramic solid-state method, and the thermal treatments were performed in air atmosphere at 1100 °C temperature. The X-ray diffraction studies have confirmed the phase composition of samples and scanning electron microscopy the shape of the crystallites has been evidenced. The perovskite material was subjected to representative magnetic investigations in order to highlight substitutions characteristics. Investigations on electrical properties have evidenced the substitution dependence of relative permittivity and electrical resistivity under humidity influence and the characteristics of humidity sensors based on this material. The results are discussed in term of microstructural changes induced by the substitutions degree and its sensor applicative effects.

## 1. Introduction

The AlFeO_3_ multiferroic perovskite is a significant ceramic material which has excellent crystal structure following the chemical formula ABO_3_ used for varied electronic and magnetic applications. This kind of multiferroic perovskite presents attractive physical properties: high value of electrical resistivity, low value of relative permittivity, low value of saturation magnetization, low production cost and good chemical stability. Further, in the last period the perovskite oxides have been receiving considerable attention because of their potential applications in electronic, non-volatile access memories, magnetic switching circuits, luminescence properties, humidity sensors, gas sensors, including microwave devices [1,2,3,4,5,6].

The main disadvantage in obtaining perovskite materials is their high sintering temperature, over 1200 °C, when material becomes more dense and large energy consumption take place. The increase of porosity degree of multiferroic perovskite materials is an unresolved problem due to the difficult volatilization of metallic cations in the A-sites positions of the crystal lattice. Partial substitution of Al^3+^ cations (covalent radius = 1.18 Å) with Sb^3+^ cations (covalent radius = 1.40 Å) and good chemical homogeneity in AlFeO_3_ perovskite can reduce the sintering temperature to about 1100 °C and can enhance the electrical and humidity sensing properties. The presence of humidity vapors in air can modify the perovskite’s resistivity by roughly two or three orders of magnitude. Various specialized studies communicate that the realization of substitutions in the A-sites position utilizing various metallic cations is a key factor that affects the physical properties of perovskite, leading to the creation of vacancies in its structure [7,8,9,10].

The fundamental aspect of perovskite humidity sensors is the dependence of humidity characteristic upon the porosity and intrinsic resistivity of the semiconductor material used, which can produce important changes in the electronic conduction mechanism [11,12]. The detection and control, as precisely as possible, of the humidity level in the environment has led to the development of a variety of materials with applications in the field of humidity sensors. Controlling the technological process and selecting the type of cations to be substituted in the perovskite structure are found to be the most important factors for achieving a high sensitivity to humidity. However, the relationship between the structure, intrinsic resistivity of the multiferroic perovskite and the humidity sensitivity is not fully understood.

Specialized studies on multiferroic perovskite ceramic materials report a great number of compounds used as sensitive elements for massive sensors [13], light sensors [14] and films [15,16], which report good response times. It is also reported in the literature that the effect of niobium [17], chromium [18] and indium [19] substitutions has been studied by many researchers in order to find the optimal composition of the multiferroic perovskite type which has spectacular electrical properties.

In the present study we investigate the effects of Sb^3+^ cations, which partially or totally substitute Al^3+^ cations in an AlFeO_3_ multiferroic perovskite, on the structure, magnetic and electrical properties. The purpose of this research is to investigate and clarify the role of the substitution of Al^3+^ ions in the AlFeO_3_ perovskite structure with Sb^3+^ ions in obtaining an active material for humidity sensors. The composition of the investigated samples is Al_1−x_Sb_x_FeO_3_ with values for x = 0.0, 0.25, 0.50, 0.75 and 1.00. In the following paragraphs we offer details about the synthesis and results of physical parameters investigation in case of Al_1−x_Sb_x_FeO_3_ multiferroic perovskite with Sb^3+^ cations substitution.

## 2. Experimental

The preparation method plays a key role in obtaining multiferroic perovskites and, due to the applicability of perovskites in various fields, in recent years various methods of their preparation have been developed, such as: high energy ball milling method [20], solid state reaction method [21], simple hydrothermal method [22] and coprecipitation method [23,24]. The Al_1−x_Sb_x_FeO_3_ multiferroic perovskite samples were prepared by following the solid-state reaction method in air atmosphere, without precautions. In order to establish the best technological parameters of synthesis and to follow the substitution influence of Sb^3+^ metal cations on the magnetic, electrical and structural properties of multiferroic perovskite, the following substitutions were made: x = 0.00, 0.25, 0.50, 0.75 and 1.00.

The raw materials used for synthesis the samples were the following: Al_2_O_3_ (Sigma-Aldrich, Germany purity ≥ 99.9%), Sb_2_O_3_ (Sigma-Aldrich, purity ≥ 99.999%) and Fe_2_O_3_ (Sigma-Aldrich, purity ≥ 99.98%). Each composition was mixed using planetary zirconium balls, Retsch model PM100, for 8 h at speeds of 500 rot/min. Calcination was performed at 500 °C temperature for 4 h and speed cooling was slowed with the furnace after disconnecting from the 380 V mains. The resulting powders were uniaxially pressed into a disc shape (pellets) with a diameter of 6 mm and a thickness of about 1.2 mm, using a pressing-down force of 3 tons using a Carver model 4350 hydraulic press. The torus shape samples (13 mm outside diameter, 9 mm inside diameter, 4 mm thick) were pressed using pressing applied force of 2 tons to investigate the magnetic properties. The compact samples, disc shape and torus shape were sintered in air at 1100 °C for 6 h with a heating rate of 5 °C/min, use a furnace Carbolite model RHF 15/8. In order to avoid the occurrence of thermal shocks or thermal stresses, the samples were cooled slowly with the furnace at a cooling rate speed 3 °C/min. To ensure the properties to be measured correspond to the solid material and not to the surface layer, for both parallel faces of all the disc samples, silver type sandwich electrodes were deposited. The structural properties of the samples were analyzed at room temperature using an X-ray diffractometer (Shimadzu LabX XRD-6000, Japan, using Bragg-Brentano geometry) with CuKα radiation (the X-ray wavelength α = 1.5405 Å), scanning electron microscopy (SEM QUANTA 200, Japan) and 3D optical surface profilometer model Zygo ZeGage. The determination of the values of some important intrinsic parameters was carried out as follows: the analysis of SEM micrographs—using the linear interception method, the average size of the crystallites was determined; analysis of XRD diffractograms—the network parameter that contributed to determining the porosity was determined; from the mass and geometric dimensions of each sample the density was determined. The specific saturation magnetization was measured use a vibrating sample magnetometer in a field of 1000 A/m on spheres type samples. The relative magnetic permeability was measured by using an inductance bridge on the torus samples. In order to be used as possible applications in the field of humidity sensors, multiferroic perovskite samples were exposed to different levels of humidity to determine the sensitivity and response time. The sample is fixed in the experimental device, in a closed enclosure, of known humidity, at the fixed temperature of 25 °C. The device is provided with 2 contact electrodes between which to insert the sample, and the change in electrical resistance or capacitance is measured. Relationships between humidity and electrical resistance at 25 °C were determined using a test chamber.

## 3. Results and Discussion

In the following the effect of Sb^3+^ substitution in the AlFeO_3_ perovskite is analyzed in terms of structure evolution, sintering behavior, magnetic and dielectric changes and sensitivity of samples to water vapor.

### 3.1. Microstructure Characteristics

The morphology and structure are two essential parameters for optimizing the properties of interest in various applications of perovskite materials. Because the degree of substitution can lead to improvements in one or more physical properties, but can also diminish other properties, is necessary to analyze the evolution of the intrinsic parameters given in Table 1.

After completing sintering treatment of samples, specific surface area (S_spec_), porosity (Φ), weight loss (Δm/m_o_), volume shrinkage (ΔV/V_o_), average grain size (D_m_) and bulk density (ρ) were determined. The parameters were obtained from analysis of SEM micrographs (for crystallites characteristics), XRD diffractograms (for the network parameter that contributed to determining the porosity), mass and geometric dimensions of each sample (for density) etc. The analysis of the values of the intrinsic parameters given in Table 1 show that in the case of Al_1−x_Sb_x_FeO_3_ multiferroic perovskite substitution of aluminium cations with antimony cations make smaller the loss of oxygen and favors the increase of the density of the samples. Moreover, from the analysis of the data presented in Table 1 it is observed that this substitution is accompanied by the decrease of the porosity, due to a reduction in the number of open pores and the increased dimensions of the grain size, therefore the densification of the perovskite material.

It is known that the magnetic and electrical properties of the multiferroic perovskite’s intrinsic parameters can be influenced by a number of factors such as: shape and size of crystallites, cationic and anionic vacancies, chemical neomogenities, dislocations, structure of granular interfaces etc. The relevant microstructural parameters considered for the analyzed samples are the bulk density (ρ) which was determined from the dimensions and weight of each sample, and the average grain size (D_m_) which was determined by the linear intercept technique to choose a number of 20 crystallites for each sample.

Figure 1 shows the dependence of some intrinsic parameters of multiferroic perovskite given by Sb^3+^ substitution, such as: density, volume shrinkage and weight loss. We found that by partially substituting Al^3+^ cations with Sb^3+^ cations for x ≥ 0.5 values there is a significant increase of volume shrinkage (ΔV/V_o_) and a slight decrease in the weight loss (Δm/m_o_).

The analysis of the phases for multiferroic perovskite was performed in air, at room temperature, using the Shimadzu model LabX-6000 diffractometer and the wavelength CuKα (the X-ray wavelength α = 1.5405 Å). Diffraction lines belonging to the perovskite type phase were identified. It is known that the degree of replacement of Al^3+^ cations by other cations in the perovskite lattice depends on the chemical–physical parameters of the substituent. It was found that the presence of a concentration of Sb^3+^ ions in the perovskite structure implies a decrease in the lattice parameter, favoring an increase in the density of the material. From the comparative analysis of the diffractograms shown in Figure 2, the presence of secondary phases is not remarked. This confirms that the sintering temperature (1100 °C) and duration (6 h) were sufficient for the formation of the perovskite lattice. Similar results are reported in the literature by [25,26,27].

The effect by partially or totally replacing Al^3+^ cations by Sb^3+^ cations in the Al_1−x_Sb_x_FeO_3_ multiferroic perovskite is shown in Figure 3. The SEM micrographs, shown in Figure 3, were performed on fractures of the samples, using a scanning electron microscope SEM model Quanta 200, in order to observe the morphology, shape and tendency of agglomeration of crystallites. It is known that the oxidation process is much more evidenced for fine-grained crystallites, due to the ratio between specific surface area and volume. From the SEM micrographs showed in Figure 3, it is evident that the structure of each sample is characterized by a distinctive intergranular porosity. Substitution of Al^3+^ cations with Sb^3+^ cations in the perovskite structure, determine to the crystallites slightly increase in size from 0.7 μm (x = 0) to 0.9 μm (x = 1).

From the micrographs shown in Figure 3, it is obvious that the sample for x = 0 shows a fine hexagonal micronic structure, constituted of small crystallites, and sample for x = 1 shows large conglomerates of crystallites placed side by side together. This structure shape indicates that the sample x = 1 can easily permit absorption and condensation of water vapor inside. Similar results are reported in the literature by [28,29].

The 3D topographic profile of the surface of the multiferroic perovskite sample Al_0.5_Sb_0.5_FeO_3_ was made using an optical profilometer model Zygo ZeGage with non-contact cantilever in normal temperature and pressure conditions. Figure 4 shows the representative parameters of the sample surface area, calculated by profilometer software: R_a_—represents the arithmetic area; S_a_—represents the mathematical area of the pixels; S_q_—represents the square area of the pixels and S_z_—represents the topographic difference between the highest and lowest points. From analysis of the surface topography of the perovskite sample shown in Figure 4, we notice that the surface is not rough and in some places presents agglomerations of crystallites, which is agreement with the SEM micrographs shown in Figure 3.

### 3.2. Magnetic Properties

The investigation of the magnetic properties was performed on sphere-shaped samples with a diameter of about 5–6 mm. Magnetic properties of the multiferroic perovskite samples were examined using Vibrating Sample Magnetometer (VSM) under influence of magnetic field in the range of −1000–1000 A/m at normal room temperature conditions. Figure 5 presents the behavior of samples magnetization as a function of the applied field. Magnetic investigations confirm that all the samples investigated belong to the category of soft magnetic materials, perovskites, because they show values close to zero for the remanence and coercivity. We remark, from graphical representation, that the magnetisation depends on the degree of substitution of aluminium cations with antimony cations. The fluctuations of the magnetic parameters, mention to Table 2, may be ascribed to the level of substitution and produced by the presence of some nonmagnetic substituent atoms in the host perovskite lattice. All the perovskite samples exhibit hysteresis behavior in the M-H curve, which indicates these are ferromagnetic. A similar result was reported in the literature by [30].

In the case of saturation magnetization (M_S_) and remanent magnetization (M_R_), the optimal substitution with Sb^3+^ cations is performed for the sample x = 0.25, where maximum values are observed (see Table 2 and Figure 5). The addition of Sb^3+^ cations in the Al_1−x_Sb_x_FeO_3_ perovskite structure manifests itself, from the point of view of the ferromagnetic properties, by decreasing the coercive field of the perovskite, which indicates that it can be used in the field of magnetic shielding, or in the field of magnetic recording.

The values shown in Table 2 indicate a slight decrease in coercivity, due to the increase in Sb^3+^ cation content in the structure of multiferroic perovskite. Similar magnetic characteristics were reported in the literature for closely related materials [31,32].

### 3.3. Electrical Properties

In order to study the electrical behavior of perovskite material it is very important to know the temperature dependence of electrical resistivity and permittivity. The electrical investigations were performed in the temperature range 200–400 K, and resistivity highlights the semiconductor behavior of the Al_1−x_Sb_x_FeO_3_ perovskite. Figure 6 shows the temperature dependence of the electrical resistivity (Figure 6a) and temperature dependence of the relative dielectric permittivity (Figure 6b) in case of multiferroic perovskite. Temperature variation measurements of electrical resistivity and relative permittivity were performed at constant frequency of 20 Hz. From the graphical representation shown in Figure 6 we found that the temperature characteristics of Al_1−x_Sb_x_FeO_3_ perovskite depend upon the intrinsic conductivity of material and microstructure. The graphical representations show a decrease up to four orders of magnitude of the electrical resistivity with temperature variation (Figure 6a) and also an increase of the dielectric permittivity with temperature variation (Figure 6b). The decrease of electrical resistivity with temperature may be due to the activation of the thermal mobility of electrons and is not thermally generated. The conduction mechanism in the perovskite materials is a typical one of semiconductors directly achieved through n-type conduction, through free electrons in the network structure and indirectly through p-type conduction. Because the samples investigations were carried out in alternating field, the majority conduction mechanism is due to a hopping process from perovskite charge carriers between the nearest neighbor atoms.

The investigations performed from the electrical point of view allow us to conclude that the introduction in the host perovskite lattice small amounts of Sb^3+^ cations, (x < 0.5) has the effect slight decrease of the electrical resistivity with temperature variation, due to the limited homogeneity. A significant amount of Sb^3+^ cations substitution in the host perovskite lattice (x ≥ 0.5) implies an increase in electrical resistivity with temperature variation. It is also noticed that, for x > 0.5 substitutions with Sb^3+^ cations in the host perovskite lattice, there is a very small variation of the electrical relative permittivity with temperature gradient. From the representations shown in Figure 6 we can conclude that the influence of Sb^3+^ cation substitution on the Al_1−x_Sb_x_FeO_3_ perovskite is only partially due to the nonhomogeneity of the mixture, and the antimony cations introduced in the host perovskite lattice significantly change the dielectric properties for substitutions x ≥ 0.5. Similar results were obtained and reported in the literature by [10,33,34,35].

The dielectric measurements shown in Figure 7, Figure 8 and Figure 9 were performed at room temperature in alternative current regime, using a LCR meter model Wayne Kerr 6400P in the frequencies ranges 20 Hz–20 MHz. It is known that the electrical resistivity of perovskites is governed mainly by the chemical composition, microstructure and electrical resistivity of crystallites conglomerations. As can be shown in Figure 7a we found that in the absence of humidity all the investigated samples show electrical resistivity variation with the frequency typical of perovskites. Moreover, all of the samples present a decrease in electrical resistivity values as the frequency increases, this being typically perovskite material behavior. We find the maximum value of electrical resistivity is obtained for x = 0.75 and the minimum value of electrical resistivity is obtained for x = 0.25. Furthermore, from graphical representations we found, for all investigated samples to the frequency range, there is an important decrease of electrical resistivity by three up four orders of magnitude.

The permittivity dependence on frequency for all samples is given in Figure 7b, and shows typical perovskite relaxation behavior, according with the Maxell–Wagner and Debye mechanism. All of the samples show an exponentially drop on the relative permittivity with the frequency increase, being in conformity by the Maxell–Wagner and Debye mechanism. The graphical representations show that, in case of sample x = 0.25, the partial substitution of Al^3+^ cations with Sb^3+^ cations involve to a slight increase in the value of relative permittivity, which is due to the reduction of electronic exchange between Fe^2+^ and Fe^3+^ ions located in octahedral positions of the perovskite structure. Furthermore, decreased value of relative permittivity with increased content of Sb^3+^ cations from Al_1−x_Sb_x_FeO_3_ perovskite is also attributed to the increase the crystallites size, because larger-size crystallites involve a smaller number of grain boundaries. Comparable results are also reported in literature [8,36].

### 3.4. Humidity Influence on the Electrical Propertiesand Humidity Sensors Characteristics

The material electrical characteristics investigations in the presence humidity were carried out at constant temperature 25 °C, using a closed enclosure of known humidity level. The perovskite electrical properties stability under humidity influence was investigated. It is known that the humidity sensitivity characteristics depend upon the intrinsic conductivity and substitution degree. Variation of resistivity and electrical capacitance as a function of the substitution degree and under humidity influence are shown in Figure 8. The highest electrical resistivity value of 10^7^ Ω·m is given by the sample with x = 0.75, evidence that electrical charges conduction mechanism in case of perovskites is dependent on microstructure. In the case of sample x = 0.75 the substitution of Al^3+^ cations by Sb^3+^ cations involves a significant increase value of electrical resistivity, by about two orders of magnitude (from 1.4 × 10^5^ to 1.2 × 10^7^ Ω·m) compared with the sample for x = 0. It can also be seen from the graphical representation shows in Figure 8a that Al_1−x_Sb_x_FeO_3_ perovskite exhibits typical semiconducting behavior in the presence of humidity, being significantly sensitive for high relative humidity levels above 65%.

It is known that the electric permittivity of a perovskite is mainly governed by the microstructure and the electrical capacitance of the grain boundaries. We found the highest value of electrical capacity is in the case of sample x = 0.75, possibly due to a homogeneous distribution of grain sizes. From the plot shown in Figure 8b it can be seen that Al_1−x_Sb_x_FeO_3_ perovskite exhibits typical perovskite dielectric behavior in the presence of humidity, with an important increase seen over 65% humidity levels.

Investigations of resistive and capacitive response time variation in relative humidity range between 0–97%RH are plotted in Figure 9. The figure shows the resistive and capacitive humidity sensitivity characteristics for Al_1−x_Sb_x_FeO_3_ perovskite sensor. As can be seen from Figure 9a for low humidity levels of about 35% the resistive coefficient R/R_0_ indicate good sensitivity, and for the high humidity levels above 75% it indicates reduced sensitivity. The decrease in the value of the resistive coefficient can be attributed to the morphology structure of the perovskite material (crystallites size and porosity). For the graphical representation given in Figure 9b, we found for low humidity levels of about 43% the capacitive coefficient C/C_0_ indicate low sensitivity, and for the high humidity levels above 75% it exponentially increases and indicates good sensitivity.

For Al_1−x_Sb_x_FeO_3_ perovskite, the response time characteristic was investigated by changing the humidity level and we found to be dependent on microstructure, as well as on degree of substitution. The investigations revealed the electrical resistivity of the samples is most influenced by their crystallite size and porosity of material.

From Figure 10a it can be noted that all samples have good acceptable response time, compared to the response time of other sensors reported in the literature [37]. In case of samples x = 0.75 and x = 1.00 for a time interval of about 200 s, the electrical resistivity value changes by two orders of magnitude when the humidity changes from 0 to 97%RH. Moreover, for values of substitution x < 0.75 although there is a decrease of electrical resistivity at humidity change, it is lower compared to samples for x ≥ 0.75. It is observed that about 75% of the total decrease of electrical resistivity occurs in less than 140 s. It seems that this substantial change in electrical resistivity has its cause in the porosity and degree of homogeneity of the perovskite. Experimentally we found that the humidity sensitivity depends on the substitution degree of the perovskite. This result indicates that this perovskite can be used for making humidity sensors. From the graphic shown in Figure 10b, a significant variation of relative permittivity with time (about 200 s) is observed for sample x = 0.75. In the case of the other samples, a smaller variation of dielectric permittivity is observed for the same time interval compared to the sample x = 0.75. A possible increase of pore size could decrease the value of response time, but this would be affecting the humidity range to which perovskite is sensitive. Next studies will focus primarily on improving Al_1−x_Sb_x_FeO_3_ perovskite response time.

## 4. Conclusions

The study of Al_1−x_Sb_x_FeO_3_ perovskite highlighted the versatility of Al substitution with Sb for improve the magnetically and electrical properties with applicative characteristics. In the case of 75%, Al^3+^ cations substitution with Sb^3+^ cations the diffusion at atomic level was highlighted and the intrinsic parameters which involved significant changes of the electrical resistivity increased by two orders of magnitude compared to the sample without substitution.

The composition and crystallites shape and size were analyzed through X-ray diffraction, electron microscopy and profilometry investigations, which highlighted the fact that the substitution of 50% and 75% with Sb^3+^ cations influences the atomic diffusion and dynamics of the sintering process causing an increase of grain size. In the case of Al_1−x_Sb_x_FeO_3_ perovskite, it has been found that the substitution of Al^3+^ cations with Sb^3+^ cations induces a densification of the perovskite, a result which is explained by the acceleration of the cationic diffusion process.

An increase of remanence and saturation magnetization in case of sample x = 0.25 was found, compared with the reference sample. The value of the coercive field decreases as the increases content of Sb^3+^ in structure of perovskite.

Humidity sensors’ applicative characteristics were identified for the sample x = 0.75 which presents an increase of the electrical resistivity by approximately two orders of magnitude compared of sample without substitution (x = 0), the presence in an important proportion of Sb favoring the structuring of these characteristics.

## Figures and Tables

**Figure 1 materials-15-08369-f001:**
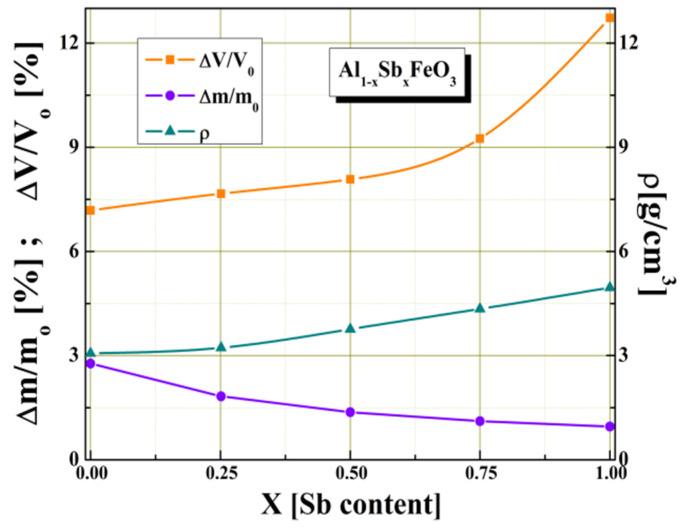
The Sb^3+^ cations content variation of some structural parameters.

**Figure 2 materials-15-08369-f002:**
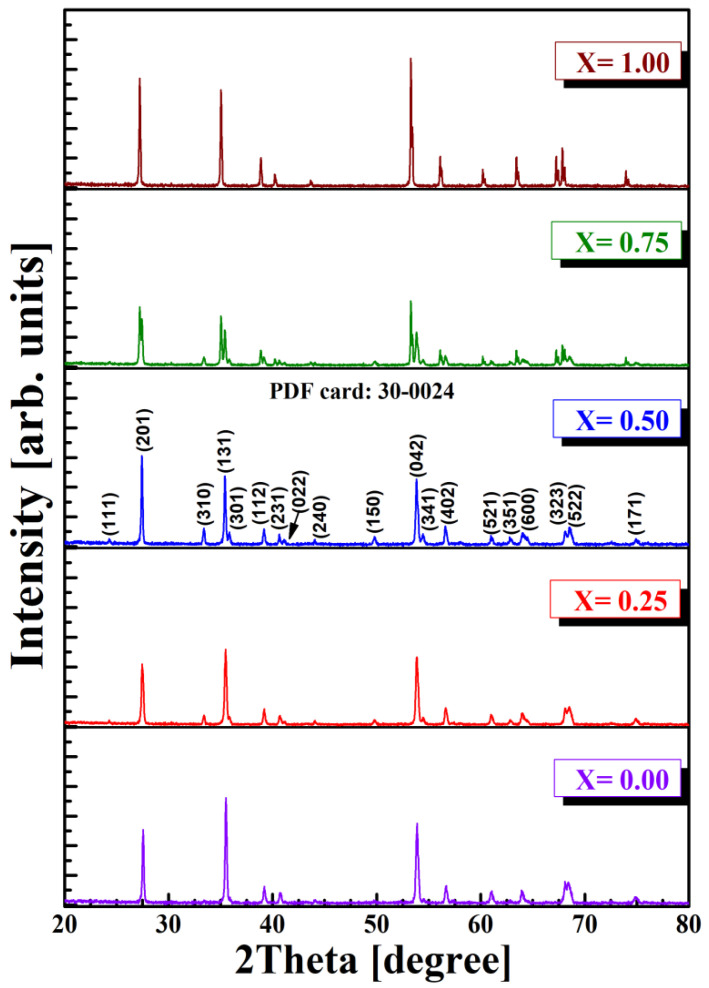
XRD patterns of Al_1−x_Sb_x_FeO_3_ perovskites.

**Figure 3 materials-15-08369-f003:**
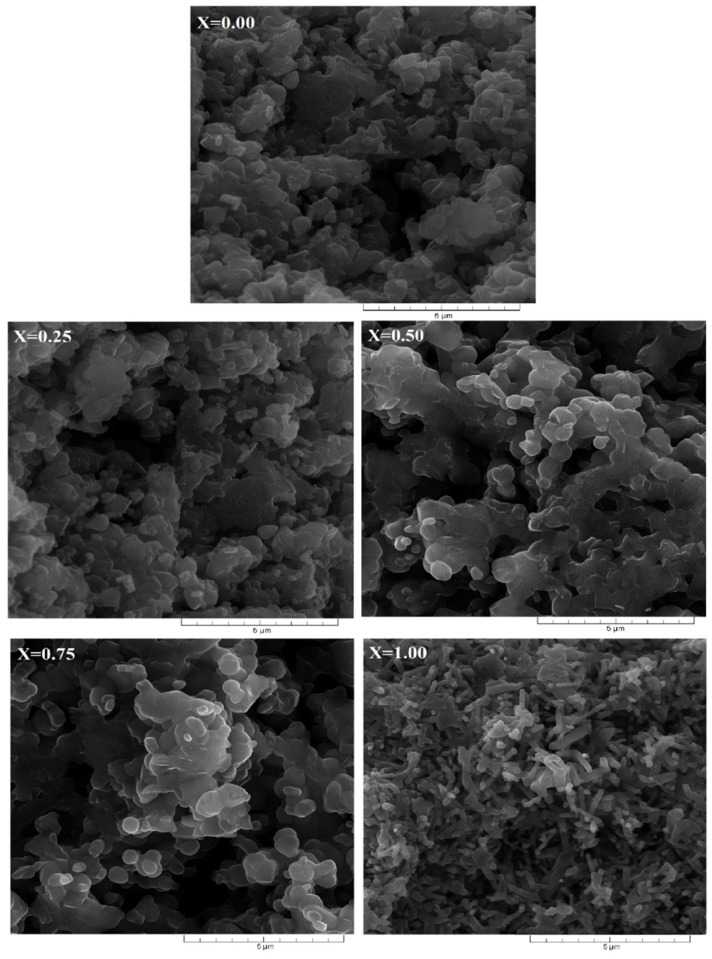
SEM micrographs of Al_1−x_Sb_x_FeO_3_ perovskites.

**Figure 4 materials-15-08369-f004:**
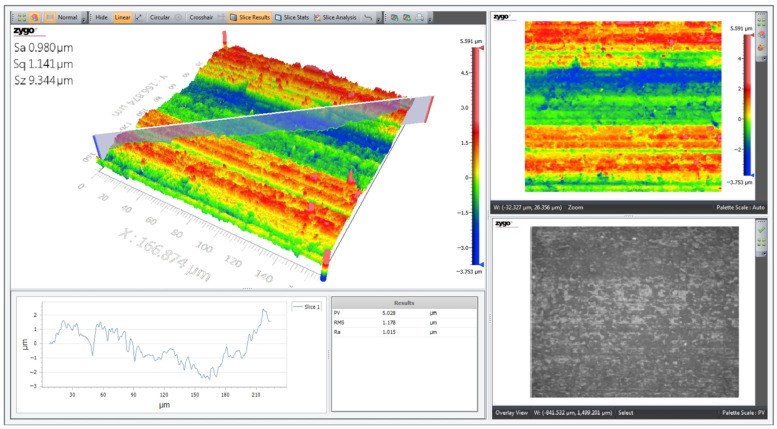
The 3D surface profile of Al_0.5_Sb_0.5_FeO_3_ multiferroic perovskite.

**Figure 5 materials-15-08369-f005:**
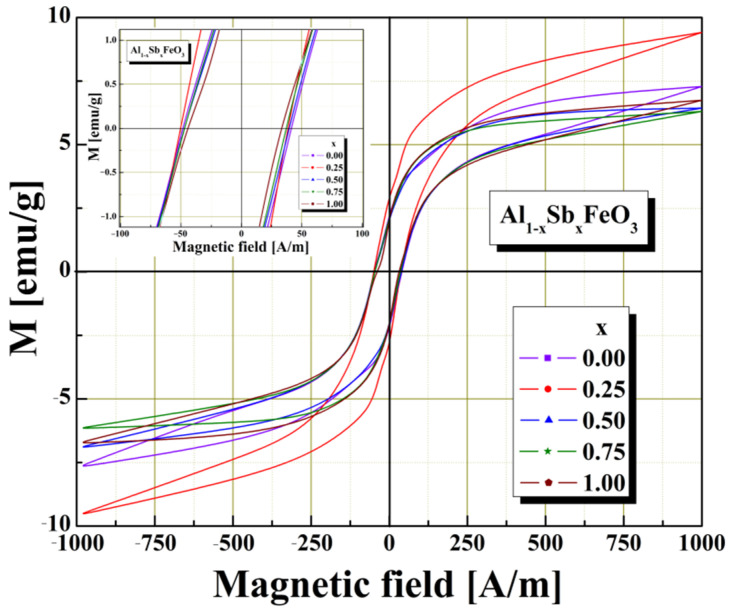
Magnetic hysteresis loops obtained for investigated multiferroic perovskite samples.

**Figure 6 materials-15-08369-f006:**
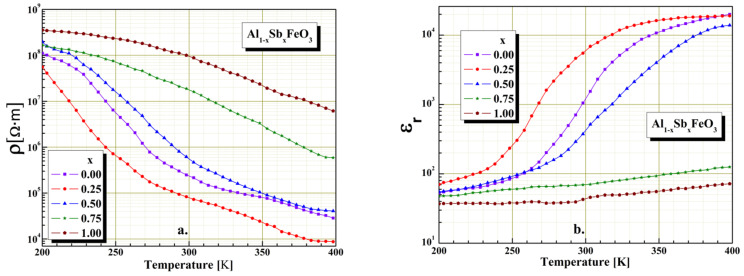
The temperature influence on electrical resistivity (**a**) and relative permittivity (**b**) at 20Hz fixed frequency for Al_1−x_Sb_x_FeO_3_ perovskites.

**Figure 7 materials-15-08369-f007:**
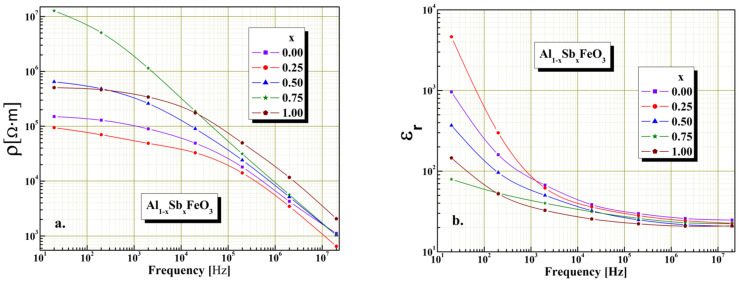
The frequency influence on electrical resistivity (**a**) and relative permittivity (**b**) without humidity for Al_1−x_Sb_x_FeO_3_ perovskites.

**Figure 8 materials-15-08369-f008:**
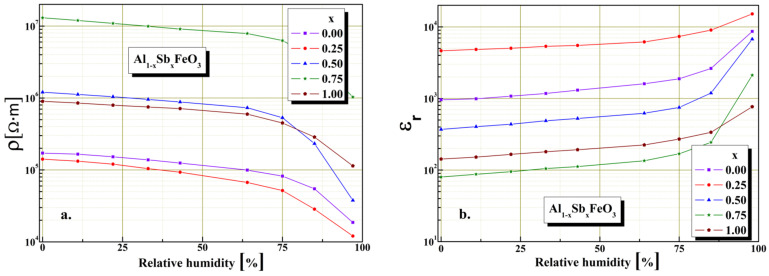
Humidity characterization: (**a**) resistive sensor and (**b**) capacitive sensor for Al_1−x_Sb_x_FeO_3_ perovskites.

**Figure 9 materials-15-08369-f009:**
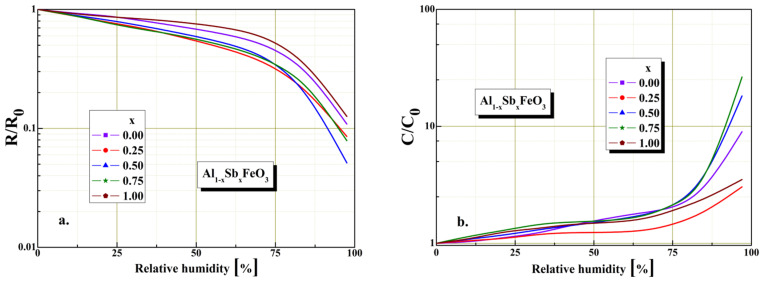
Humidity sensitivity of resistive and capacitive sensors based on Al_1−x_Sb_x_FeO_3_ perovskite material: (**a**) electrical resistance in normalized form (**b**) electrical capacitance in normalized form.

**Figure 10 materials-15-08369-f010:**
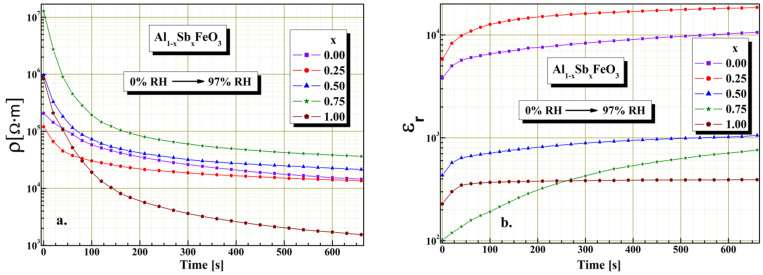
Response time for resistive sensor (**a**) and capacitive sensor (**b**) for Al_1−x_Sb_x_FeO_3_ perovskites.

**Table 1 materials-15-08369-t001:** Evolution of some intrinsic parameters for Al_1−x_Sb_x_FeO_3_ perovskite.

Sample x	S_spec_ [mm^2^]	Porosity Φ [%]	D_m_ [μm]	Bulk Density ρ [g/cm^3^]
0.00	2.706	35.42	0.724	3.063
0.25	2.408	33.50	0.773	3.224
0.50	1.970	32.68	0.810	3.759
0.75	1.634	31.45	0.845	4.345
1.00	1.301	31.12	0.930	4.956

**Table 2 materials-15-08369-t002:** Summary of the magnetic properties of Al_1−x_Sb_x_FeO_3_ perovskites.

Sample x	Saturation M_S_ [emu/g]	Remanence M_R_ [emu/g]	Coercivity H_C_ [A/m]
0.00	7.30	2.26	41.35
0.25	9.35	3.11	39.78
0.50	6.37	2.12	38.72
0.75	6.25	2.17	37.17
1.00	6.69	2.19	33.54

## Data Availability

Not applicable.

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
