# Peer review of "Influence of Sb3+ Cations on the Structural, Magnetic and Electrical Properties of AlFeO3 Multiferroic Perovskite with Humidity Sensors Applicative Characteristics"

_materials, 2022, doi:10.3390/ma15238369_

Round 1

Reviewer 1 Report

I find that the manuscript entitled: "Influence of Sb3+ cations on the structural, magnetic and electrical properties of AlFeO3 multiferroic perovskite with humidity sensors applicative characteristics" is an interesting.

The paper deals with an analysis of the effects of Sb3+ cations substitution on the structural, magnetic and electrical properties of Al1-xSbxFeO3 multiferroic perovskite are investigated. In paper was presented the partially or totally substitution of Al3+ cations with Sb3+ cations, in stoichiometric composition Al1-xSbxFeO3 (x=0.00; 0.25; 0.50; 0.75 and 1.00) were made in order to identify composite materials with sensors applicative properties. After the thermal treatments in air atmosphere at 1100 oC, characterization of obtained materials was done by the X-ray diffraction and scanning electron microscopy, as well as by the investigation of the magnetic and electrical properties. Investigations on electrical properties have evidenced the substitution dependence of relative permittivity and electrical resistivity under humidity influence and the characteristics of humidity sensors based on synthesized materials.

However, the work has certain drawbacks:

  1. Table 1 presents the parameters of samples: specific surface area (Sspec), porosity (Φ), weight loss (Δm/mo), volume shrinkage (ΔV/Vo), average grain size (Dm) and bulk density (ρ), but methodology was not given for them in the experimental part.
  2. 2. In the Introduction, the authors mentioned the values of the covalent radius for Al3+ and Sd3+, but did not present the values of the lattice parameters obtained by using the XRD method, as well as the influence of the replacement of the mentioned ions and their covalent radii on the stability of the material structure.
  3. 3. The authors talk about increasing the conductivity of the material, but in the experimental part they did not specify the methodology and model of the device they used for electrical measurements. Also, in the presentation of the results, they refer to the activation of the thermal mobility of electrons, which they claim is not thermally generated. On what basis do the authors claim that the carriers of conductivity are only electrons and not ions?
  4. 4. The authors presented the results of humidity influence on the electrical properties and humidity sensors characteristics, but there is no description of a clear methodology for performing the measurements in the experiment. They also state that electrical measurements are made at a constant temperature, but it doesn't state which one? It is also not clear how to measure electrical conductivity at temperature in a chamber with changing humidity? I ask the authors to explain it to the experimental part as well?

I recommend the authors to review the manuscript and eliminate all shortcomings, in order to make a scientific contribution to the sent paper.

Accordingly, I recommend the accept after minor revision (corrections to minor methodological errors and text editing).

Reviewer 2 Report

By synthesizing the Sb-doped AlFeO3 samples, the authors characterized the structure using XRD and investigated the magnetic, electrical properties and the characteristics of humidity sensors. It is interesting to see the compositional dependence of these properties. However, the following issues should be considered to further revise manuscript.

1.     The authors should give a detailed figure caption, for each of them, for readers to better understand the results. As for Fig. 4, what kind of core information was provided herein to the entire research?

2.     From Fig. 2, one can see that the XRD patterns change as a function of the Sb doping concentration, especially for the case of x = 0.75 and 1.00 against others. The authors should specify how many different phases coexist in each sample. What are the structural and/or secondary phases? e.g., is there Al2O3, Sb2O3 and iron oxides? if the XRD refinement can be done, that would be the best.

3.     The authors characterized the sample morphology using SEM. To answer one important question: whether Sb was incorporated into the AlFeO3 matrix, the elemental maps should be presented as well.

4.     In Fig. 2, the bottom tick of the figure may overlap with the diffraction peaks, leading to potential diffusion. This should be revised for a clearer presentation.

5.     Regarding the property change as a function of the Sb doping concentration, could the authors give more insights and in-depth discussion?

Overall, a major revision should be made before having a further decision.

Reviewer 3 Report

In this article, the authors investigated various characteristics of Al1xSbxFeO3 ceramic samples. The authors compared material properties by preparing different material compositions. However, there is no mechanistic explanation for the optimization results, which I think should be revised.

1.     The SEM images are too blurry. Please magnify the images to clarify them.

2.     The information in Figure 4 cannot be distinguished.

3.     In the introduction section, the significance of this research is not mentioned.

4.     Why does the substitution of Al ions will increase the grain size?

5.     In line 217, maybe an error: “200 ÷ 400 K”.

6.     Through a series of comparative experiments, the author draws the conclusion that the material has the best performance when x=0.75. This is obtained through experimental results. However, is there any theoretical rationale why 0.75 is a good choice?

Round 2

Reviewer 2 Report

Unfortunately, the authors didn't make further efforts to revise the manuscript and improve its quality in terms of the last-round review comments.This leads to pending of a number of questions and the conclusions are not solid enough.

In my view, further revision should be done for further consideration.

Reviewer 3 Report

I think this version is enough to be published.